# Transcriptomic Analysis Reveals LncRNAs Associated with Flowering of *Angelica sinensis* during Vernalization

**Xiaoxia Liu [1], Mimi Luo [1], Mengfei Li [1,\*] and Jianhe Wei [2,\*]**

[1] State Key Laboratory of Aridland Crop Science, College of Life Science and Technology, Gansu Agricultural University, Lanzhou 730070, China; 18893481962@163.com (X.L.); luomimi9521@163.com (M.L.)

[2] Institute of Medicinal Plant Development, Chinese Academy of Medical Sciences & Peking Union Medical College, Beijing 100193, China

\* Correspondence: lmf@gsau.edu.cn (M.L.); jhwei@implad.ac.cn (J.W.)

**Abstract:** *Angelica sinensis* is a "low-temperature and long-day" perennial plant that produces bioactive compounds such as phthalides, organic acids, and polysaccharides for various types of clinical agents, including those with cardio-cerebrovascular, hepatoprotective, and immunomodulatory effects. To date, the regulatory mechanism of flowering under the photoperiod has been revealed, while the regulatory network of flowering genes during vernalization, especially in the role of lncRNAs, has yet to be identified. Here, lncRNAs associated with flowering were identified based on the full-length transcriptomic analysis of *A. sinensis* at vernalization and freezing temperatures, and the coexpressed mRNAs of lncRNAs were validated by qRT-PCR. We obtained a total of 2327 lncRNAs after assessing the protein-coding potential of coexpressed mRNAs, with 607 lncRNAs aligned against the TAIR database of model plant *Arabidopsis*, 345 lncRNAs identified, and 272 lncRNAs characterized on the SwissProt database. Based on the biological functions of coexpressed mRNAs, the 272 lncRNAs were divided into six categories: (1) chromatin, DNA/RNA and protein modification; (2) flowering; (3) stress response; (4) metabolism; (5) bio-signaling; and (6) energy and transport. The differential expression levels of representatively coexpressed mRNAs were almost consistent with the flowering of *A. sinensis*. It can be concluded that the flowering of *A. sinensis* is positively or negatively regulated by lncRNAs, which provides new insights into the regulation mechanism of the flowering of *A. sinensis*.

**Keywords:** *Angelica sinensis*; flowering; lncRNA; vernalization; transcriptomic analysis

## 1. Introduction

*Angelica sinensis* (Oliv.) Diels is a "low-temperature and long-day" perennial plant that is native to Gansu Province, Northwest China [1,2]. The roots have been used as a traditional Chinese medicine for over 2000 years [3,4]. Currently, the roots are also applied in cardio-cerebrovascular, hepatoprotective, and immunomodulatory clinical agents, largely relying on bioactive compounds such as phthalides, organic acids, and polysaccharides [5–7].

Recently, the planting area of *A. sinensis* has exceeded 40,000 ha to satisfy the increasing demand; however, a higher rate (>40%) of early bolting and flowering (EBF) in commercial cultivation increases the lignified rate of roots and decreases the yield accordingly [8,9]. In order to inhibit the EBF, effective measures that have been taken include selecting excellent germplasm resources with a lower rate of EBF [8], controlling the seedling size (0.4 to 0.6 cm) to delay the transition from vegetative to reproductive growth [10], storing seedlings below freezing temperatures (−3 to −10 °C) to avoid vernalization (0 to 10 °C) [2,11,12], and shading the plants with sunshade nets (40% to 60%) to avoid the long-day conditions during the adult stages [13].

Regarding the regulatory mechanism of flowering in *A. sinensis*, key genes and the regulatory network during the photoperiod have been identified and mapped based on transcriptomic analysis. Specifically, 13 genes associated with the photoperiod, vernalization,

sucrose, and GA pathways were identified from plants at the vegetative stage compared with the EBF stage [14]; 38 genes associated with the photoperiod, carbohydrates, hormone signaling, and floral development were identified from different development stages [15]; and 40 genes associated with the photoperiod, sucrose, GA, and floral development were identified from the EBF compared with Un-EBF [16]. In summary, key genes such as *FLC*, *SOC1*, *FT*, *PHYA*, *AP1*, and $GA_2OX_1$ were identified during the transition from the vegetative to the reproductive stage [14–16]. To date, physiological changes in the levels of carbohydrates, proteins, and hormones during vernalization have been investigated [17,18], while the regulatory network of flowering has yet to be identified.

Generally, ncRNAs include two categories, housekeeping and regulatory ncRNAs, and the latter can be further divided into sRNAs (i.e., miRNA, siRNA, and piRNA) and lncRNAs (>200 nucleotides) [19]. Both miRNAs and lncRNAs can influence plant developmental processes and stress responses [20], with the former being negative regulators functioning as specificity determinants, or guides, within complexes that promote the degradation of mRNA targets, and the latter acting either as precursors of miRNAs or endogenous target mimics (TMs), which mimic the real targets of miRNAs, thus rendering the corresponding miRNAs ineffective [21]. For example, previous studies on resistance against leaf rust in wheat found that 50 miRNAs and 1178 lncRNAs were identified and 49 lncRNAs were found to be the targets for miRNAs, with 1 lncRNA acting as a precursor of 2 miRNAs, and 3 lncRNAs acting as TMs [22].

Extensive investigations have demonstrated that lncRNAs regulate their downstream targets' expression through the changing of epigenetic modification at the level of transcription and post-transcription by interacting with DNA, RNA, and proteins; thus, they are involved in various biological processes [23–25]. In this study, lncRNAs associated with flowering were identified based on the transcriptomic analysis of *A. sinensis* seedlings treated at vernalization and freezing temperatures (avoiding vernalization). We found that 272 lncRNAs directly or indirectly participate in regulating the flowering of *A. sinensis* and stress responses.

## 2. Materials and Methods

### 2.1. Plant Material

The seedlings (root–shoulder diameter 0.4–0.5 cm; Figure S1) of *Angelica sinensis* (cultivar Mingui 1) were stored at 0 (vernalization temperature) and −3 °C (freezing temperature), respectively. After storage at 0 °C for 14 (T1) and 60 days (T2), as well as at −3 °C for 125 days (T3), the shoot apical meristem (SAM) was cut from the root shoulder of the seedlings for transcriptomic analysis and qRT-PCR validation. Three biological replicates were performed for each treatment of T1, T2, and T3. Herein, the treatment of T1, T2, and T3 represents uncompleted, completed, and avoided vernalization, respectively, based on the EBF rate (Figure S2) when the stored seedlings were cultivated and grown in a long-day condition.

### 2.2. Full-Length Isoform Sequencing and Analysis

Total RNA of the SAM samples was extracted using Trizol reagent (Omega Bio-Tek, Norcross, GA, USA). The integrity of the RNA was determined using an Agilent 2100 Bioanalyzer (Agilent Technol., California, CA, USA) and agarose gel electrophoresis, and the purity and concentration of the RNA were determined using a microspectrophotometer (NanoDrop Technol., Wilmington, DE, USA). The high-quality RNAs were sequenced on a Pacific Biosciences Sequel platform (Gene Denovo Biotechnology Co., Ltd., Guangzhou, China). Raw reads of cDNA library were analyzed using a SMRT Link (V8.0.0) [26]. Briefly, high-quality CCS were extracted from the subreads BAM file; the integrity of transcripts (full-length sequences) was judged based on whether CCS reads contained primers (5′ and 3′) and polyAs; then, FLNC reads were generated by removing primers, barcodes, and polyAs; finally, FLNC reads were assembled to obtain the entire isoform [27].

### 2.3. Analysis of Long Noncoding RNAs (lncRNAs)

Isoforms that were not annotated against the four databases—NR, Swiss-Prot, Kyoto KEGG, and KOG—were used for the analysis of lncRNAs. The isoform that was assessed as a noncoding transcript by both CNCI and CPC software was finally confirmed as a lncRNA [28,29].

### 2.4. Characterization of LncRNAs

To date, the genome of *A. sinensis* has not been sequenced. Thus, the lncRNA analysis of *A. sinensis* was performed via a BLAST search with an E-value cut-off of $\leq 1 \times 10^{-5}$ against the known lncRNAs from the TAIR database (https://www.arabidopsis.org accessed on 30 March 2022) [30]. The function of lncRNAs was annotated based on their coexpression mRNAs [31–33]. Herein, the biological functions of the coexpressed mRNAs were searched on the UniProt database (https://www.uniprot.org accessed on 30 March 2022).

### 2.5. qRT-PCR Validation

Based on the coding sequences (CDS) of coexpressed mRNA of lncRNA, 49 primer sequences of representatively coexpressed mRNAs (Table 1) were designed using the NCBI primer-blast tool. First-strand cDNA synthesis and qRT-PCR reaction were carried out using SuperRealPreMix Plus (FP205; Tiangen Biotech., Beijing, China) according to the manufacturer's instructions; specifically, the cDNA was synthesized successively with one cycle (95 °C, 15 min) and 40 cycles (95 °C, 10 s; 60 °C, 20 s; and 72 °C, 30 s), and the qRT-PCR reaction was incubated successively at 95 °C for 15 s, 60 °C for 1 min and 95 °C for 1 s. The *Actin* (*ACT*) gene was used as a reference control gene with forward: TGGTATTGTGCTGGATTCTGGT and reverse: TGAGATCACCACCAGCAAGG (amplicon size 109 bp) [34]. Herein, the cycle threshold (Ct) values and standard curves of the ACT gene at different volumes (0.25, 0.5, 1.0, 1.5, 2.0, and 3.0 µL) was built to correct the gene expression level (Figure S3 and Figure S4), and the expression levels of the 49 candidate genes and their standard deviations for every variant were added to the Supplementary Materials (Table S1). The REL of coexpressed mRNAs was calculated using the $2^{-\triangle\triangle Ct}$ method [35] according to the following formula.

$$\triangle Ct_{\text{Test gene}} = Ct_{\text{Test gene}} - Ct_{\text{Reference gene}}$$

$$\triangle Ct_{\text{Target gene}} = Ct_{\text{Target gene}} - Ct_{\text{Reference gene}}$$

$$-\triangle\triangle Ct_{(\text{T2 vs. T1})} = -(\triangle Ct_{\text{T2}} - \triangle Ct_{\text{T1}})$$

$$-\triangle\triangle Ct_{(\text{T3 vs. T1})} = -(\triangle Ct_{\text{T3}} - \triangle Ct_{\text{T1}})$$

Relative expression level (REL) = $2^{-\triangle\triangle Ct}$.

**Table 1.** Primer sequences used in qRT-PCR validation.

| lncRNA ID | Coexpressed mRNAs | mRNA ID | Primer Sequences (5′ to 3′) | Amplicon Size (bp) |
|---|---|---|---|---|
| Isoform0062250 | *HMGB2* | NM_001035997.1 | Forward: CAAAGCTGCTGCTAAGGAC | 155 |
| | | | Reverse: GGACTTCCACTTGTCTCCAGC | |
| Isoform0061796 | *HMGB3* | NM_001035998.1 | Forward: CCTTCCAGTGCCTTCTTCGT | 174 |
| | | | Reverse: CTCAACCTTGCGCTTGTCAG | |
| Isoform0001498 | *At1g05910* | NM_100472.2 | Forward: AGACCACTCTCTCCGGTTGT | 109 |
| | | | Reverse: TCGTCAACTCCGATGACGTG | |
| Isoform0062769 | *RID2* | NM_125110.6 | Forward: GCAGGGCTTAGGTCTTCGTT | 100 |
| | | | Reverse: ACGAGGTTCATGCGATGACT | |

**Table 1.** *Cont.*

| lncRNA ID | Coexpressed mRNAs | mRNA ID | Primer Sequences (5′ to 3′) | Amplicon Size (bp) |
|---|---|---|---|---|
| Isoform0061049 | *At4g26600* | NM_001341820.1 | Forward: TTCCGATTGGTGCAACTCCT | 118 |
| | | | Reverse: GCCATGTCCACAACTCGTTC | |
| Isoform0034756 | *H2AV* | NM_001339683.1 | Forward: CAGTTGGACGAATTCACAGGC | 176 |
| | | | Reverse: CAGATGCCTTGGCGTTATCC | |
| Isoform0050517 | *At2g28720* | NM_128433.4 | Forward: GCAAGAAGCTTCAAAATTAGC | 107 |
| | | | Reverse: TGCTTAGCAAGTTCACCAGG | |
| Isoform0062503 | *HTR2* | NM_113651.2 | Forward: CACCGGAGGAGTGAAGAAGC | 189 |
| | | | Reverse: TCTTGAAGAGCTGCGACTGC | |
| Isoform0027210 | *HOS15* | NM_126132.4 | Forward: TACAGGCGCAGAACCTATGG | 164 |
| | | | Reverse: CTGTTGCATCACCAGACCCT | |
| Isoform0062048 | *REF6* | NM_148863.4 | Forward: AGGGAACACAGCTTCTGGTG | 124 |
| | | | Reverse: TTCCCCAAGTGAACGGTCTG | |
| Isoform0062818 | *SKP1A* | NM_106245.5 | Forward: GTGCTGCTACCTCCGATGAC | 181 |
| | | | Reverse: GTGCGGATCTCTTCTGGAGT | |
| Isoform0061474 | *ASK21* | NM_001125404.1 | Forward: CCTGATGACCTTACTGAGGAG | 178 |
| | | | Reverse: CAGGTCATCCACTGAACGCT | |
| Isoform0061497 | *SRK2G* | NM_120946.5 | Forward: ACATCGAGAGAGGTCGCAAG | 110 |
| | | | Reverse: AGGTGTCAGGATCACCTCCTT | |
| Isoform0062575 | *SRK2H* | NM_125760.2 | Forward: TGGTCCTGTGGTGTGACTCT | 164 |
| | | | Reverse: GAGAGAAGGTGTCTGCACTCC | |
| Isoform0062220 | *SRR1* | NM_125348.4 | Forward: ATCGCATTGTTTGGGAACAGC | 117 |
| | | | Reverse: AGCAAACTCGCTTGTGACTCT | |
| Isoform0061284 | *PHL* | NM_001334526.1 | Forward: CAAAGTCCTCGTTTGTCGGC | 104 |
| | | | Reverse: GCAACTGCTCCATAGTGGGT | |
| Isoform0057927 | *PHYA* | NM_001123784.1 | Forward: GTGCGATATGCTGATGCGTG | 149 |
| | | | Reverse: CCTGCAGGTGGAACTCACTT | |
| Isoform0041956 | *AGL62* | NM_125437.5 | Forward: CTCCTCACCAACACAACAAC | 197 |
| | | | Reverse: AACGCAAGTTCCTCAACGGG | |
| Isoform0045502 | *AGL79* | NM_113925.3 | Forward: AATCACCCCATGAGCTTCGC | 107 |
| | | | Reverse: TAGGGTTCCGGCAGCTACTT | |
| Isoform0063170 | *ATJ3* | NM_114279.4 | Forward: GAATACGCTCACGGAGTTGC | 135 |
| | | | Reverse: GCATCCCACTTGGCTCTCTC | |
| Isoform0062470 | *ACBP6* | NM_102916.4 | Forward: AATCACCCCATGAGCTTCGC | 107 |
| | | | Reverse: TAGGGTTCCGGCAGCTACTT | |
| Isoform0061783 | *ENO2* | NM_129209.4 | Forward: CACTGAGTGTGGAACCGAGG | 190 |
| | | | Reverse: GGTCATCACTCCCCAACCTG | |
| Isoform0063049 | *ADH1* | NM_106362.3 | Forward: TGTGACCGAGTGTGTGAACC | 123 |
| | | | Reverse: TGAATCATGGCCTGAACGCT | |
| Isoform0062198 | *CSP2* | NM_120029.3 | Forward: GATCTGGAGGTGGATACGGC | 115 |
| | | | Reverse: CAGTCTCTCGCCATGTGACC | |

**Table 1.** *Cont.*

| lncRNA ID | Coexpressed mRNAs | mRNA ID | Primer Sequences (5′ to 3′) | Amplicon Size (bp) |
|---|---|---|---|---|
| Isoform0062617 | *HSP17.8* | NM_100614.3 | Forward: AACATCGGCGATAACGAACG | 154 |
| | | | Reverse: CTCCACGTGTCTCTCTCCAC | |
| Isoform0009507 | *HSP70-3* | NM_001202918.1 | Forward: CGACTGCAGGAGACACTCAT | 144 |
| | | | Reverse: TCTCACAGGCGGTTCTCAAC | |
| Isoform0034676 | *HSP70-10* | NM_120996.4 | Forward: CGTGTCCCCAAGGTTCAGTC | 167 |
| | | | Reverse: CCGAGCGATAGAGGTGTGAC | |
| Isoform0042993 | *HSP90-3* | NM_124983.4 | Forward: AACAAGGAGGAGTACGCTGC | 193 |
| | | | Reverse: AGACACGACGGACATAGAGC | |
| Isoform0061974 | *CSY4* | NM_001337082.1 | Forward: GATGCAGAGCTCTACCGACC | 197 |
| | | | Reverse: CCTCTTCCGGGTCAAGCAAT | |
| Isoform0062375 | *GAPCP2* | NM_101496.3 | Forward: CATTTCTGCACCTTCAGCGG | 198 |
| | | | Reverse: TTCTGAGTAGCTGTGGTCGC | |
| Isoform0062268 | *At3g52990* | NM_115159.5 | Forward: GACAACTTGCGACCAACTCG | 101 |
| | | | Reverse: AATCCACGAAGGGTCTCAGC | |
| Isoform0062585 | *PGM2* | NM_001160993.2 | Forward: TGAACTGCGTACCCAAGGAG | 158 |
| | | | Reverse: TCGGTCTGCATCACCATCAG | |
| Isoform0062370 | *BAM1* | NM_113297.3 | Forward: AACTCTCTCGCTGTTCCTCG | 165 |
| | | | Reverse: GGAGAAGCCCGTCTCACAAT | |
| Isoform0062586 | *ARF1* | NM_001337250.1 | Forward: GTGACCGTGTTGTTGAAGCC | 148 |
| | | | Reverse: TGAAGCCCAAGCTTGTCAGT | |
| Isoform0061377 | *CUL1* | NM_001036498.3 | Forward: GTGCCGTGCATTGCTAAGAG | 153 |
| | | | Reverse: TCTTCGGCCTGTTGGACAAG | |
| Isoform0057235 | *SOFL4* | NM_123240.2 | Forward: AGGTCGTGGATGAGGACTAC | 144 |
| | | | Reverse: GAACCGCTGATAATTTGGCCC | |
| Isoform0043114 | *SOFL5* | NM_001342234.1 | Forward: TGCGAGTCAGGATGGACTCT | 193 |
| | | | Reverse: TCCTTGGACCAGAAGAAGCAT | |
| Isoform0062152 | *GRF2* | NM_106479.3 | Forward: AACTCTCCGGAATCTGCGAC | 192 |
| | | | Reverse: GAGCAGATTTGTAAGCGGCG | |
| Isoform0062828 | *GRF11* | NM_001084180.2 | Forward: GGTGCTAGGAGAGCATCGTG | 198 |
| | | | Reverse: GACGGTGGATTCTCCCGAAG | |
| Isoform0061395 | *ERF3* | NM_103946.3 | Forward: ATCGTTTAGCGGACCCAGAC | 101 |
| | | | Reverse: CGCAATCGCTGTGACAATCC | |
| Isoform0022533 | *SF1* | NM_001036978.2 | Forward: GGCTTAGGGTCAACTCCGAC | 164 |
| | | | Reverse: CCAGTCACACGGTCCTTGAT | |
| Isoform0044730 | *PURU1* | NM_124115.4 | Forward: ACGTCTTCTACTCTCGCAGC | 125 |
| | | | Reverse: AGGCACACGCACAACTGAAT | |
| Isoform0047216 | *MES16* | NM_117770.5 | Forward: CCATCCCTTCTCCGCATCTT | 195 |
| | | | Reverse: TCATAGGAGCAGGACGCAAC | |
| Isoform0063248 | *GPT1* | NM_124861.5 | Forward: CGCTGGTTCGTTGATGATGC | 193 |
| | | | Reverse: AAACGCAGGTTCACCACTCT | |

**Table 1.** *Cont.*

| lncRNA ID | Coexpressed mRNAs | mRNA ID | Primer Sequences (5′ to 3′) | Amplicon Size (bp) |
|---|---|---|---|---|
| Isoform0062571 | *ABCF5* | NM_125882.3 | Forward: TGCTGATAGGCTTGTGGCTT | 103 |
| | | | Reverse: CGGCTCATCAAGTAGCAGCA | |
| Isoform0008194 | *SECA2* | NM_001198130.1 | Forward: ACTGTGAGGCCCATTGTCTG | 117 |
| | | | Reverse: CTCTGCCACGAAGCTGGTTA | |
| Isoform0062373 | *VPS26A* | NM_124733.4 | Forward: TGTTCCGCTTCCTCCAATCAA | 196 |
| | | | Reverse: TGCTCCAGTTGATTCTCGCC | |
| Isoform0062449 | *CML19* | NM_119864.5 | Forward: CGAGCTCAACGTTGCTATGAG | 160 |
| | | | Reverse: GTCTATGGAGTCTCGTTCTCCG | |
| Isoform0061307 | *NHX6* | NM_106609.4 | Forward: GGCATTTGCTCTTGCTCTGC | 112 |
| | | | Reverse: TCCTCCAATCAGCAACACCG | |

*2.6. Statistical Analysis*

In order to obtain the precise estimation of PCR efficiency, each experiment for qRT-PCR validation was performed with three biological replicates, along with three technical replicates [36]. A t-test in SPSS 22.0 was performed for independent experiments, with $p < 0.05$ as the basis for statistical differences.

### 3. Results

*3.1. LncRNAs Analysis*

In total, 2327 lncRNAs were obtained after assessing the protein-coding potential of coexpressed mRNAs based on the two software programs, CNCI and CPC (Figure 1A), with 607 genes aligned against the known lncRNAs from the TAIR database of model plant *Arabidopsis* (Figure 1B), and 345 lncRNAs with coexpressed mRNAs of *A. sinensis* identified (Figure 1C) based on the SwissProt database. Based on the biological functions, the 272 characterized lncRNAs (Figure 1D) were divided into six categories: chromatin, DNA/RNA and protein modification (29); flowering (36); stress response (24); metabolism (117); biosignaling (23), and energy and transport (43) (Figure 1E). The base sequences of the 272 lncRNAs are shown in Table S2.

*3.2. LncRNAs Linked with Chromatin, DNA/RNA and Protein Modification, as well as Expression Levels of Their Coexpressed mRNAs*

Based on the biological functions of coexpressed mRNAs, 29 lncRNAs were linked with chromatin (*HMGB2* and *HMGB3*), DNA/RNA (*At1g05910*, *RID2* and *At4g26600*) and protein modification (*H2AV*, *At2g28720*, *HTR2*, *HOS15*, *REF6*, *SKP1A*, *ASK21*, *SRK2G*, *SRK2H*, *DET1*, *BOPAt4g295601*, *At1g45180*, *At3g50840*, *At2g36630*, *At3g47890*, *UBP7*, *DER2.1*, *GRP3*, *MDH9.13*, *At3g24715*, *At3g16560*, *At3g62260*, *ESMD1*, and *At1g27930*) (Table 2). The expression levels of 14 representative coexpressed mRNAs were confirmed by qRT-PCR, with 3 mRNAs (HMGB2, HMGB3 and At1g05910) showing down-regulation at T2 versus (vs.) T1, and 11 mRNAs showing lower levels at T2 vs. T1 than T3 vs. T1, with the exception of 2 mRNAs (H2AV and ASK21) (Figure 2).

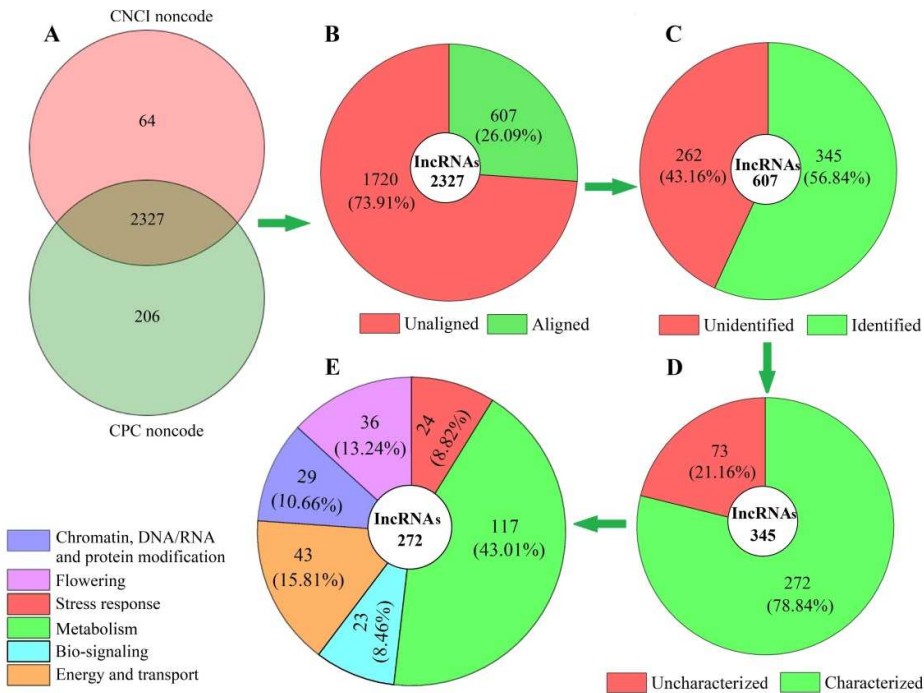

**Figure 1.** Distribution and classification of lncRNAs in *Angelica sinensis* during vernalization, based on the biological functions of coexpressed mRNAs. Abbreviations: CNCI, coding–noncoding index; CPC, coding potential calculator. Images (**A**), (**B**), (**C**), (**D**) and (**E**) represent total, aligned, identified, characterized and classified lncRNAs, respectively.

**Table 2.** Twenty-nine lncRNAs linked with chromatin, DNA/RNA, and protein modification.

| lncRNA ID | Coexpressed mRNAs | mRNA ID | Proteins Encoded by Coexpressed mRNAs |
|---|---|---|---|
| **Chromatin modification (2)** | | | |
| Isoform0062250 | *HMGB2* | AT1G20693.1 | High mobility group B protein 2 |
| Isoform0061796 | *HMGB3* | AT1G20696.1 | High mobility group B protein 3 |
| **DNA/RNA modification (3)** | | | |
| Isoform0001498 | *At1g05910* | AT1G05910.1 | ATPase family AAA domain-containing protein At1g05910 |
| Isoform0062769 | *RID2* | AT5G57280.1 | 18S rRNA (guanine-N(7))-methyltransferase RID2 e |
| Isoform0061049 | *At4g26600* | AT4G26600.8 | S-adenosyl-L-methionine-dependent methyltransferases superfamily protein |
| **Protein modification (24)** | | | |
| Isoform0034756 | *H2AV* | AT3G54560.1 | Histone H2A variant 1 |
| Isoform0050517 | *At2g28720* | AT2G28720.1 | Histone H2B.3 |
| Isoform0062503 | *HTR2* | AT1G09200.1 | Histone H3.2 |
| Isoform0027210 | *HOS15* | AT5G67320.1 | WD40 repeat-containing protein HOS15 |
| Isoform0062048 | *REF6* | AT3G48430.1 | Lysine-specific demethylase REF6 |
| Isoform0062818 | *SKP1A* | AT1G75950.1 | SKP1-like protein 1A |
| Isoform0061474 | *ASK21* | AT3G61415.1 | SKP1-like protein 21 |
| Isoform0061497 | *SRK2G* | AT5G08590.1 | Serine/threonine-protein kinase SRK2G |
| Isoform0062575 | *SRK2H* | AT5G63650.1 | Serine/threonine-protein kinase SRK2H |
| Isoform0053138 | *DET1* | AT4G10180.1 | Light-mediated development protein DET1 |
| Isoform0030135 | *BOPAt4g295601* | AT3G57130.2 | Ankyrin repeat family protein/BTB/POZ domain-containing protein |
| Isoform0063715 | *At1g45180* | AT1G45180.1 | F27F5.26 |
| Isoform0044146 | *At3g50840* | AT3G50840.3 | Phototropic-responsive NPH3 family protein |
| Isoform0034163 | *At2g36630* | AT2G36630.1 | Sulfite exporter TauE/SafE family protein 4 |
| Isoform0058941 | *At3g47890* | AT3G47890.2 | Ubiquitin carboxyl-terminal hydrolase-related protein |
| Isoform0007659 | *UBP7* | A0A1I9LL79 | Ubiquitin carboxyl-terminal hydrolase |

**Table 2.** *Cont.*

| lncRNA ID | Coexpressed mRNAs | mRNA ID | Proteins Encoded by Coexpressed mRNAs |
|---|---|---|---|
| Isoform0045576 | *DER2.1* | AT3G21280.2 | Derlin-2.1 |
| Isoform0062510 | GRP3 | AT2G05520.1 | Glycine-rich protein 3 |
| Isoform0062575 | *MDH9.13* | AT5G42440.1 | At5g42440 |
| Isoform0051826 | *At3g24715* | AT3G24715.1 | Kinase superfamily with octicosapeptide/Phox/Bem1p domain-containing protein |
| Isoform0061548 | *At3g16560* | AT3G16560.4 | Probable protein phosphatase 2C 40 |
| Isoform0049189 | *At3g62260* | AT3G62260.1 | Probable protein phosphatase 2C 49 |
| Isoform0060667 | *ESMD1* | AT2G01480.1 | Protein ESMERALDA 1 |
| Isoform0062775 | *At1g27930* | AT1G27930.1 | Probable methyltransferase At1g27930 |

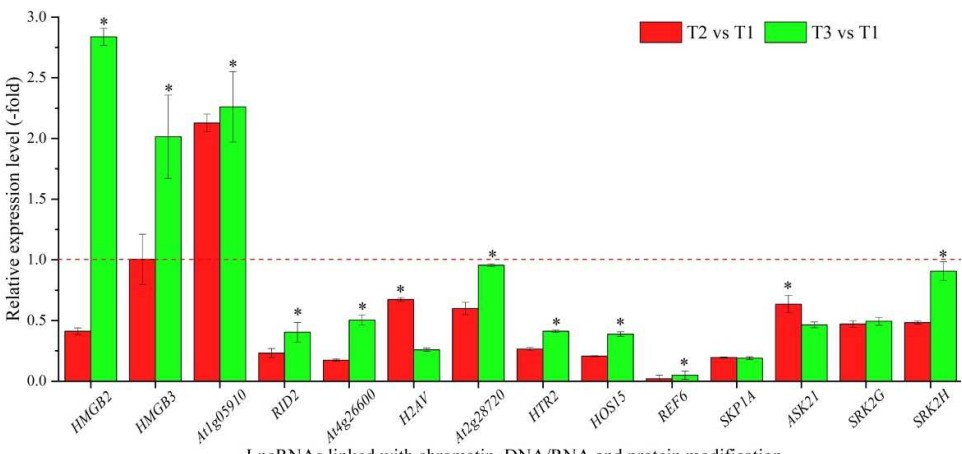

**Figure 2.** The expression levels of coexpressed mRNAs of lncRNAs linked with chromatin, DNA/RNA and protein modification in *A. sinensis* at T2 vs. T1 and T3 vs. T1, as determined by qRT-PCR. T1, T2, and T3 represent uncompleted, completed, and avoided vernalization, respectively. The "*" represents a significant difference at $p < 0.05$ level between T2 vs. T1 and T3 vs. T2 for the same gene.

### 3.3. LncRNAs Linked with Flowering and Expression Levels of Their Coexpressed mRNAs

In total, 36 lncRNAs were linked with flowering based on the biological functions of their coexpressed mRNAs, with 12 lncRNAs directly associated with flowering, namely *SRR1*, *PHL*, *PHYA*, *AGL62*, *AGL79*, *ATJ3*, *BBX29*, *CLE13*, *CLE44*, *MXC17.10*, *At1g06515*, and *BHLH30* (Table 3), and 24 lncRNAs indirectly associated with flowering, such as cell division, embryo development, and cell wall organization (Table S3). The expression levels of six representative coexpressed mRNAs (*SRR1*, *PHL*, *PHYA*, *AGL62*, *AGL79*, and *ATJ3*) were confirmed by qRT-PCR, with all six mRNAs showing down-regulation at T2 vs. T1 and T3 vs. T1, and five mRNAs showing lower levels at T2 vs. T1 than T3 vs. T1, with the exception of the gene *AGL62* (Figure 3).

### 3.4. LncRNAs Linked with Stress Response and Expression Levels of Their Coexpressed mRNAs

In total, 24 lncRNAs were linked with the stress response based on the biological functions of their coexpressed mRNAs, with 14 lncRNAs associated with the temperature response, namely *ACBP6*, *ENO2*, *ADH1*, *CSP2*, *RH20*, *RH52*, *RH53*, *RAB18*, *XERO1*, *MED14*, *HSP17.8*, *HSP70-3*, *HSP70-10*, and *HSP90-3* (Table 4), and 10 lncRNAs associated with other stresses responses, such as water, salt, and oxidative stress (Table S4). The expression levels of eight representative coexpressed mRNAs involved in the temperature response were confirmed by qRT-PCR, with four mRNAs (*ACBP6*, *ENO2*, *CSP2*, and *HSP90-3*) showing up-regulation at T3 vs. T1, and three mRNAs (*ACBP6*, *ENO2*, and *CSP2*) showing lower levels at T2 vs. T1 than T3 vs. T1 (Figure 4).

**Table 3.** Twelve lncRNAs directly linked with flowering.

| lncRNA ID | Coexpressed mRNAs | mRNA ID | Proteins Encoded by Coexpressed mRNAs |
|---|---|---|---|
| Isoform0062220 | *SRR1* | AT5G59560.2 | Protein SENSITIVITY TO RED LIGHT REDUCED 1 |
| Isoform0061284 | *PHL* | AT1G72390.1 | Protein PHYTOCHROME-DEPENDENT LATE-FLOWERING |
| Isoform0057927 | *PHYA* | AT1G09570.6 | Phytochrome A |
| Isoform0041956 | *AGL62* | AT5G60440.1 | Agamous-like MADS-box protein AGL62 |
| Isoform0045502 | *AGL79* | AT3G30260.1 | AGAMOUS-like 79 |
| Isoform0063170 | *ATJ3* | AT3G44110.1 | Chaperone protein dnaJ 3 |
| Isoform0028325 | *BBX29* | AT5G54470.1 | At5g54470 |
| Isoform0054894 | *CLE13* | AT1G73965.1 | CLAVATA3/ESR (CLE)-related protein 13 |
| Isoform0061298 | *CLE44* | AT4G13195.1 | CLAVATA3/ESR (CLE)-related protein 44 |
| Isoform0056751 | *MXC17.10* | AT5G24710.1 | Transducin/WD40 repeatlike superfamily protein |
| Isoform0062524 | *At1g06515* | AT1G06515.2 | Transmembrane protein, putative (DUF3317) |
| Isoform0061573 | *BHLH30* | AT1G68810.1 | Transcription factor bHLH30 |

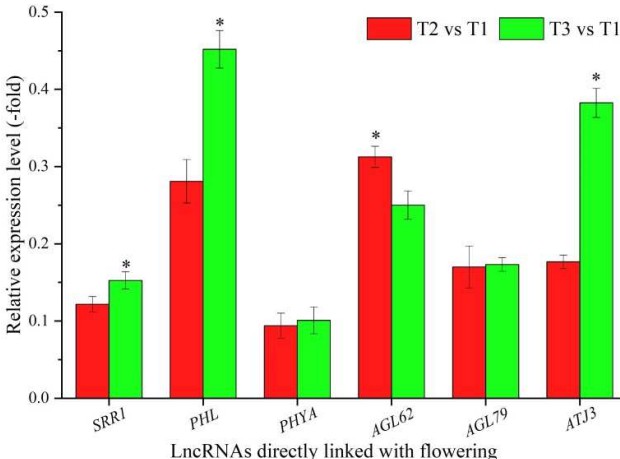

**Figure 3.** The expression levels of coexpressed mRNAs of lncRNAs directly linked with flowering in *A. sinensis* at T2 vs. T1 and T3 vs. T1, as determined by qRT-PCR. The "*" represents a significant difference at $p < 0.05$ level between T2 vs. T1 and T3 vs. T2 for the same gene.

**Table 4.** Fourteen lncRNAs directly linked with temperature response.

| lncRNA ID | Coexpressed mRNAs | mRNA ID | Proteins Encoded by Coexpressed mRNAs |
|---|---|---|---|
| Isoform0062470 | *ACBP6* | AT1G31812.1 | Acyl-CoA-binding domain-containing protein 6 |
| Isoform0061783 | *ENO2* | AT2G36530.1 | Bifunctional enolase 2/transcriptional activator |
| Isoform0063049 | *ADH1* | AT1G77120.1 | Alcohol dehydrogenase class-P |
| Isoform0062198 | *CSP2* | AT4G38680.1 | Cold shock protein 2 |
| Isoform0035932 | *RH20* | AT1G55150.2 | DEA(D/H)-box RNA helicase family protein |
| Isoform0019851 | *RH52* | AT3G58570.1 | DEAD-box ATP-dependent RNA helicase 52 |
| Isoform0062484 | *RH53* | AT3G22330.1 | DEAD-box ATP-dependent RNA helicase 53, mitochondrial |
| Isoform0058095 | *RAB18* | AT5G66400.1 | Dehydrin Rab18 |
| Isoform0061968 | *XERO1* | AT3G50980.1 | Dehydrin Xero 1 |
| Isoform0020919 | *MED14* | AT3G04740.1 | Mediator of RNA polymerase II transcription subunit 14 |
| Isoform0062617 | *HSP17.8* | AT1G07400.1 | 17.8 kDa class I heat shock protein |
| Isoform0009507 | *HSP70-3* | AT3G09440.2 | Heat shock 70 kDa protein 3 |
| Isoform0034676 | *HSP70-10* | AT5G09590.1 | Heat shock 70 kDa protein 10, mitochondrial |
| Isoform0042993 | *HSP90-3* | AT5G56010.1 | Heat shock protein 90-3 |

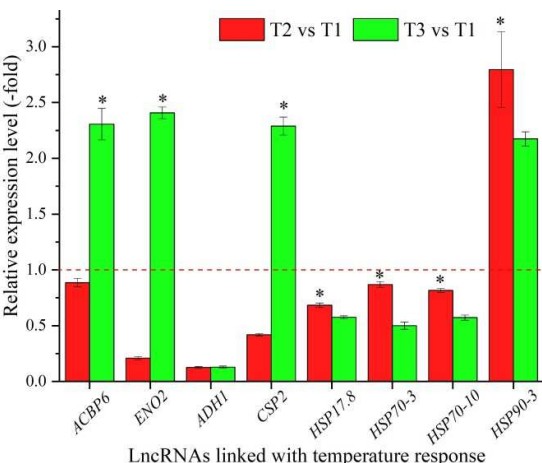

**Figure 4.** The expression levels of coexpressed mRNAs of lncRNAs linked with temperature response in *A. sinensis* at T2 vs. T1 and T3 vs. T1, as determined by qRT-PCR. The "*" represents a significant difference at $p < 0.05$ level between T2 vs. T1 and T3 vs. T2 for the same gene.

### 3.5. LncRNAs Linked with Metabolism and Expression Levels of Their Coexpressed mRNAs

In total, 117 lncRNAs were linked with metabolism based on the biological functions of their coexpressed mRNAs, with 19 lncRNAs associated with carbohydrate metabolism, namely *CSY4*, *FBA3*, *GAPC1*, *GAPCP2*, *At3g52990*, *PGM2*, *USP*, *GLCNAC1PUT2*, *UXS2*, *XYLA*, *GALS1*, *OFUT31*, *At5g67460*, *RSW3*, *At1g59950*, *At5g25970*, *UGT76E7*, *At1g26850*, and *BAM1* (Table 5), and 98 lncRNAs associated with other types of metabolism, such as nucleotide, protein, and lipid metabolism (Table S5). The expression levels of five representative coexpressed mRNAs (*CSY4*, *GAPCP2*, *At3g52990*, *PGM2*, and *BAM1*) involved in carbohydrate metabolism were confirmed by qRT-PCR, with all five mRNAs showing down-regulation at T3 vs. T1, and three mRNAs showing higher levels at T2 vs. T1 than T3 vs. T1, with the exception of the two genes *GAPCP2* and *BAM1* (Figure 5).

**Table 5.** Nineteen lncRNAs directly linked with carbohydrate metabolism.

| lncRNA ID | Coexpressed mRNAs | mRNA ID | Proteins Encoded by Coexpressed mRNAs |
|---|---|---|---|
| Isoform0061974 | *CSY4* | AT2G44350.2 | Citrate synthase 4, mitochondrial |
| Isoform0062251 | *FBA3* | AT2G01140.1 | Fructose-bisphosphate aldolase 3, chloroplastic |
| Isoform0062131 | *GAPC1* | AT3G04120.1 | Glyceraldehyde-3-phosphate dehydrogenase GAPC1, cytosolic |
| Isoform0062375 | *GAPCP2* | AT1G16300.1 | Glyceraldehyde-3-phosphate dehydrogenase GAPCP2, chloroplastic |
| Isoform0062268 | *At3g52990* | AT3G52990.1 | Pyruvate kinase |
| Isoform0062585 | *PGM2* | AT1G70730.3 | Phosphoglucomutase (alpha-D-glucose-1,6-bisphosphate-dependent) |
| Isoform0063034 | *USP* | AT5G52560.1 | UDP-sugar pyrophosphorylase |
| Isoform0042435 | *GLCNAC1PUT2* | AT2G35020.1 | UDP-N-acetylglucosamine diphosphorylase 2 |
| Isoform0062018 | *UXS2* | AT3G62830.1 | UDP-glucuronic acid decarboxylase 2 |
| Isoform0062750 | *XYLA* | AT5G57655.2 | Xylose isomerase |
| Isoform0060700 | *GALS1* | AT2G33570.1 | Galactan beta-1,4-galactosyltransferase GALS1 |
| Isoform0062032 | *OFUT31* | AT4G24530.1 | O-fucosyltransferase 31 |
| Isoform0039872 | *At5g67460* | AT5G67460.1 | Glucan endo-1,3-beta-D-glucosidase |
| Isoform0045893 | *RSW3* | AT5G63840.2 | Glycosyl hydrolases family 31 protein |
| Isoform0036200 | *At1g59950* | AT1G59950.1 | Aldo/keto reductase |
| Isoform0042053 | *At5g25970* | AT5G25970.2 | Core-2/I-branching beta-1,6-N-acetylglucosaminyl-transferase family protein |
| Isoform0059181 | *UGT76E7* | AT5G38040.1 | UDP-glycosyltransferase 76E7 |
| Isoform0037698 | *At1g26850* | AT1G26850.2 | Probable methyltransferase PMT2 |
| Isoform0062370 | *BAM1* | AT3G23920.1 | Beta-amylase 1, chloroplastic |

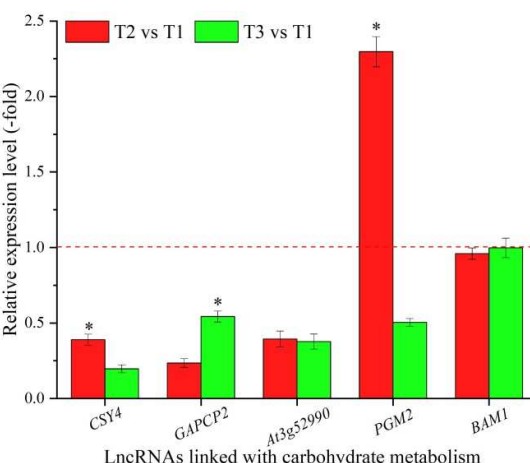

**Figure 5.** The expression levels of coexpressed mRNAs of lncRNAs linked with carbohydrate metabolism in *A. sinensis* at T2 vs. T1 and T3 vs. T1, as determined by qRT-PCR. The "*" represents a significant difference at $p < 0.05$ level between T2 vs. T1 and T3 vs. T2 for the same gene.

### 3.6. LncRNAs Linked with Biosignaling and Expression Levels of Their Coexpressed mRNAs

In total, 23 lncRNAs were linked with metabolism based on the biological functions of their coexpressed mRNAs, with 13 lncRNAs associated with hormone signaling, namely *ARF1*, *CUL1*, *T4L20.330*, *SOFL4*, *SOFL5*, *GRF2*, *GRF11*, *ERF3*, *CIPK20*, *SF1*, *AGD9*, *TIFY4B*, and *At2g34810* (Table 6), and 10 lncRNAs associated with other types of signaling, such as protein kinase, phosphatidylinositol-mediated, and cell surface receptor signaling (Table S6). The expression levels of eight representative coexpressed mRNAs associated with hormone signaling were confirmed by qRT-PCR, with three mRNAs (*ARF1*, *CUL1*, and *GRF11*) showing up-regulation at T2 vs. T1 and T3 vs. T1, and four mRNAs (*ARF1*, *SOFL4*, *GRF2*, and *ERF3*) showing lower levels at T2 vs. T1 than T3 vs. T1 (Figure 6).

**Table 6.** Thirteen lncRNAs directly linked with hormone signaling.

| lncRNA ID | Coexpressed mRNAs | mRNA ID | Proteins Encoded by Coexpressed mRNAs |
|---|---|---|---|
| Isoform0062586 | *ARF1* | AT2G47170.1 | ADP-ribosylation factor 1 |
| Isoform0061377 | *CUL1* | AT4G02570.1 | Cullin-1 |
| Isoform0015752 | *T4L20.330* | AT4G34750.1 | SAUR-like auxin-responsive protein family |
| Isoform0057235 | *SOFL4* | AT5G38790.1 | Protein SOB FIVE-LIKE 4 |
| Isoform0043114 | *SOFL5* | AT4G33800.2 | Protein SOB FIVE-LIKE 5 |
| Isoform0062152 | *GRF2* | AT1G78300.1 | 14-3-3-like protein GF14 omega |
| Isoform0062828 | *GRF11* | AT1G34760.1 | 14-3-3-like protein GF14 omicron |
| Isoform0061395 | *ERF3* | AT1G50640.1 | Ethylene-responsive transcription factor 3 |
| Isoform0007735 | *CIPK20* | AT5G45820.1 | CBL-interacting serine/threonine-protein kinase 20 |
| Isoform0022533 | *SF1* | AT5G51300.2 | Splicing factorlike protein 1 |
| Isoform0046311 | *AGD9* | AT5G46750.1 | Probable ADP-ribosylation factor GTPase-activating protein AGD9 |
| Isoform0057859 | *TIFY4B* | AT4G14720.1 | Protein TIFY 4B |
| Isoform0062437 | *At2g34810* | AT2G34810.1 | Berberine bridge enzyme-like 16 |

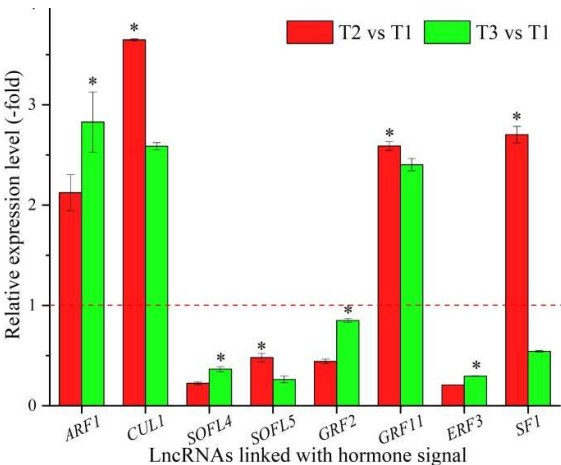

**Figure 6.** The expression levels of coexpressed mRNAs of lncRNAs linked with biosignaling in *A. sinensis* at T2 vs. T1 and T3 vs. T1, as determined by qRT-PCR. The "*" represents a significant difference at $p < 0.05$ level between T2 vs. T1 and T3 vs. T2 for the same gene.

### 3.7. LncRNAs Linked with Energy and Transport

In all, 43 lncRNAs were found to be linked with energy and transport based on the biological functions of their coexpressed mRNAs, with 5 lncRNAs associated with energy, such as *PURU1*, *ndhB1*, and *NDB1*, and 38 lncRNAs associated with transport, such as *At5g11230*, *GPT1* and *SMXL5* (Table 7). The expression levels of eight representative coexpressed mRNAs (*PURU1*, *MES16*, *GPT1*, *ABCF5*, *SECA2*, *VPS26A*, *CML19*, and *NHX6*) were confirmed by qRT-PCR, with three mRNAs (*PURU*, *GPT1*, and *VPS26A*) showing up-regulation at T2 vs. T1 and T3 vs. T1, and six mRNAs (*PURU1*, *GPT1*, *ABCF5*, *SECA2*, *VPS26A*, and *CML19*) showing higher levels at T2 vs. T1 than T3 vs. T1, with the exception of the two genes *MES16* and *NHX6* (Figure 7).

**Table 7.** Forty-three lncRNAs linked with energy and transport.

| lncRNA ID | Coexpressed mRNAs | mRNA ID | Proteins Encoded by Coexpressed mRNAs |
|---|---|---|---|
| **Energy (5)** | | | |
| Isoform0044730 | *PURU1* | AT5G47435.2 | Formyltetrahydrofolate deformylase 1, mitochondrial |
| Isoform0031698 | *ndhB1* | ATCG01250.1 | NAD(P)H-quinone oxidoreductase subunit 2 A, chloroplastic |
| Isoform0029144 | *NDB1* | AT4G28220.2 | NADH:ubiquinone reductase (nonelectrogenic) |
| Isoform0046995 | *WNK9* | AT5G28080.3 | Nonspecific serine/threonine protein kinase |
| Isoform0047216 | *MES16* | AT4G16690.1 | pFDCC methylesterase MES16 |
| **Transport (38)** | | | |
| Isoform0047603 | *At5g11230* | AT5G11230.1 | Probable sugar phosphate/phosphate translocator At5g11230 |
| Isoform0063248 | *GPT1* | AT5G54800.1 | Glucose-6-phosphate/phosphate translocator 1, chloroplastic |
| Isoform0053163 | *SMXL5* | AT5G57130.1 | Protein SMAX1-LIKE 5 |
| Isoform0062571 | *ABCF5* | AT5G64840.1 | ABC transporter F family member 5 |
| Isoform0062182 | *SFH8* | AT2G21520.2 | Phosphatidylinositol/phosphatidylcholine transfer protein SFH8 |
| Isoform0022607 | *BASS6* | AT4G22840.1 | Probable sodium/metabolite cotransporter BASS6, chloroplastic |
| Isoform0008194 | *SECA2* | AT1G21650.3 | Protein translocase subunit SECA2, chloroplastic |
| Isoform0014041 | *At4g14160* | AT4G14160.1 | Protein transport protein SEC23 |
| Isoform0062660 | *ycf2-A* | ATCG01280.1 | Protein Ycf2 |
| Isoform0036773 | *At4g22990* | AT4G22990.1 | SPX domain-containing membrane protein At4g22990 |
| Isoform0032676 | *At3g49350* | AT3G49350.1 | At3g49350 |
| Isoform0062443 | *TMT2* | AT4G35300.5 | Tonoplast monosaccharide transporter2 |
| Isoform0003274 | *VPS24-1* | AT5G22950.1 | Vacuolar protein sorting-associated protein 24 homolog 1 |
| Isoform0062373 | *VPS26A* | AT5G53530.1 | Vacuolar protein sorting-associated protein 26A |
| Isoform0028575 | *VPS52* | AT1G71270.1 | Vacuolar protein sorting-associated protein 52 A |
| Isoform0038654 | *VPS60-2* | AT5G04850.1 | Vacuolar protein sorting-associated protein 60.2 |

**Table 7.** *Cont.*

| lncRNA ID | Coexpressed mRNAs | mRNA ID | Proteins Encoded by Coexpressed mRNAs |
|---|---|---|---|
| Isoform0062996 | *At5g19500* | AT5G19500.1 | At5g19500 |
| Isoform0062800 | *PIP1-5* | AT4G23400.1 | Probable aquaporin PIP1-5 |
| Isoform0061992 | *SULTR2;2* | AT1G77990.1 | Sulfate transporter 2.2 |
| Isoform0062606 | *MT2A* | AT3G09390.2 | Metallothionein-like protein 2A |
| Isoform0057136 | *At3g52300* | AT3G52300.1 | ATP synthase subunit d, mitochondrial |
| Isoform0061450 | *ABCB23* | AT4G28630.1 | ABC transporter B family member 23, mitochondrial |
| Isoform0062449 | *CML19* | AT4G37010.1 | Calcium-binding protein CML19 |
| Isoform0061763 | *JJJ1* | AT1G74250.1 | DNAJ protein JJJ1 homolog |
| Isoform0061955 | *PAP1* | AT1G13750.1 | Probable inactive purple acid phosphatase 1 |
| Isoform0062074 | *HIPP04* | AT1G29000.2 | Heavy metal-associated isoprenylated plant protein 4 |
| Isoform0062972 | *HIPP26* | AT4G38580.1 | Heavy metal-associated isoprenylated plant protein 26 |
| Isoform0014712 | *KINUA* | AT1G12430.1 | Kinesin-like protein KIN-UA |
| Isoform0063051 | *IQD30* | AT1G18840.2 | Protein IQ-DOMAIN 30 |
| Isoform0016777 | *At3g18430* | AT3G18430.1 | Calcineurin b subunit (Protein phosphatase 2b regulatory subunit)-like protein |
| Isoform0063921 | *CBL10* | AT4G33000.6 | Calcineurin B-like protein 10 |
| Isoform0061307 | *NHX6* | AT1G79610.1 | Sodium/hydrogen exchanger 6 |
| Isoform0024682 | *CNGC15* | AT2G28260.2 | Cyclic nucleotide-gated channel 15 |
| Isoform0034638 | *2-Oct* | AT1G79360.1 | Organic cation/carnitine transporter 2 |
| Isoform0033810 | *CNGC13* | AT4G01010.2 | Putative cyclic nucleotide-gated ion channel 13 |
| Isoform0051075 | *MTG13.4* | AT5G16680.1 | RING/FYVE/PHD zinc finger superfamily protein |
| Isoform0051563 | *SUF4* | AT1G30970.3 | Zinc finger (C2H2 type) family protein |
| Isoform0037483 | *AHA4* | AT3G47950.2 | ATPase 4, plasma membranetype |

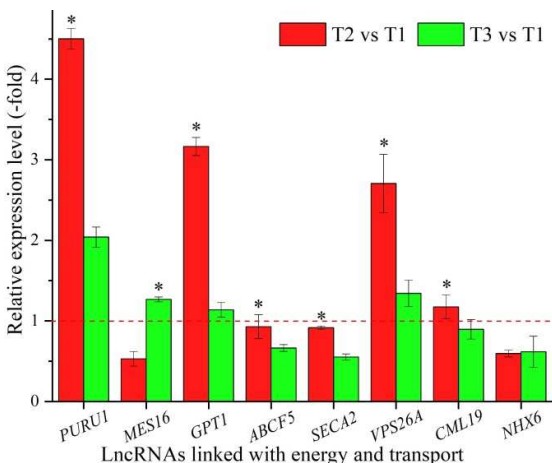

**Figure 7.** The expression levels of coexpressed mRNAs of lncRNAs linked with energy and transport in *A. sinensis* at T2 vs. T1 and T3 vs. T1, as determined by qRT-PCR. The "*" represents a significant difference at $p < 0.05$ level between T2 vs. T1 and T3 vs. T2 for the same gene.

## 4. Discussion

Vernalization is a process considered to be an epigenetic switch whereby flowering is promoted by prolonged exposure to cold (0 to 10 °C); meanwhile, it can be lost at high temperatures or avoided below freezing temperatures [37,38]. Epigenetic regulation involves diverse molecular mechanisms including chromatin remodeling, DNA methylation, histone modification, histone variants, and ncRNAs [39]. Studies on *Brassica rapa* found that 127 differentially expressed lncRNAs were coexpressed with 128 differentially expressed genes, indicating that lncRNAs played an important role during vernalization [40]. In this study, 272 characterized lncRNAs were identified from *A. sinensis* and divided into six categories, namely (1) chromatin, DNA/RNA, and protein modification; (2) flowering;

(3) stress response; (4) metabolism; (5) biosignaling; and (6) energy and transport, based on their coexpressed mRNAs.

FLC is a MADS-box transcriptional regulator that acts as a potent repressor of flowering [38]. In *Arabidopsis*, the epigenetic silencing of the floral repressor gene *FLC* is a well-characterized key event of vernalization [41]. In this study, 29 lncRNAs linked with chromatin, DNA/RNA, and protein modification were identified in *A. sinensis* during vernalization. For the chromatin modification, two coexpressed mRNAs, *HMGB2* and *HMGB3*, are involved in binding preferentially double-stranded DNA and up-regulated in response to cold stress [42]. For the DNA/RNA modification, *At1g05910* is involved in DNA demethylation and the negative regulation of chromatin silencing [43]; *RID2* is involved in rRNA methyltransferase activity [44]; and *At4g26600* is involved in RNA methylation [45]. For protein modification, 24 lncRNAs were involved and the roles of nine coexpressed mRNAs were represented as follows. *H2AV* plays a central role in regulating transcription, repairing DNA, replicating DNA, and stabilizing the nucleus chromosome [46]; *At2g28720* and *HTR2* are involved in compacting DNA into chromatin [47,48]; *HOS15* promotes the deacetylation of histone H4 [49]; *REF6* is involved in demethylating 'Lys-27' of histone H3, regulating flowering time by repressing *FLC* expression and interacting with the NF-Y complex to regulate SOC1 [50,51]; *SKP1A* and *ASK21* are involved in ubiquitination and form an SCF E3 ubiquitin ligase complex together with CUL1, RBX1, and an F-box protein [52,53]; and *SRK2G* and *SRK2H* are involved in protein phosphorylation [54,55]. In previous studies, *HMGB2* and *HMGB3* were up-regulated in response to cold stress but down-regulated in response to drought and salt stresses [42]; *H2AV*, *At2g28720* and *HTR2* exhibited a high level in response to osmotic and drought stresses [56]; *HOS15* was found to act as a repressor of cold stress-regulated gene expression and played a role in gene regulation for plant acclimation and tolerance to cold stress [49]; *SKP1A* and *ASK21* were overexpressed in the host stress response [57]; and *SRK2G* and *SRK2H* were found to be positive regulators in stress responses such as drought, salt, and cold [54,55,58]. Currently, although most of the lncRNAs have been reported to be involved in the stress response in many plants, their roles in the regulation of flowering time have been studied in model plants [23]. Previous studies have found that FLC in *Arabidopsis* is epigenetically regulated by lncRNAs *COOLAIR* and *COLDAIR* [59–61]. In this study, a lower expression level was noted for almost all coexpressed mRNAs of lncRNAs involved in chromatin, DNA/RNA, and protein modification at vernalization (T2, 0 °C) compared with freezing temperature (T3, −3 °C), as well as down-regulation at T2, indicating that these lncRNAs participate in epigenetic silencing by transferring euchromatin to heterochromatin and confer early bolting and flowering of *A. sinensis*. Representatively, the down-regulation of mRNAs *HMGB2* and *HMGB3* during vernalization transfers the heterochromatin to the euchromatin [42], which generates the ability of flowering; in contrast, their up-regulation at freezing temperatures inhibits flowering by keeping the heterochromatin, which indicates that their coexpressed lncRNAs play positive roles in regulating the flowering of *A. sinensis*. The down-regulation of mRNA *REF6* below 0 °C weakens the demethylation of histone H3 and delays flowering time by inducing FLC expression [50,51]; meanwhile, there is an increased expression level with decreased temperatures, which indicates that this coexpressed lncRNA plays a negative role in regulating the flowering of *A. sinensis*.

Flower formation occurs at the SAM and is a complex morphological event that is required not only for the circadian clock to measure the passage of time but also the regulation of meristem identity genes [42]. For the 6 representative coexpressed mRNAs of the 12 lncRNAs directly linked with flowering, *SRR1* is involved in a circadian clock input pathway and regulation of the expression of clock-regulated genes such as *CCA1* and *TOC1* [62]; *PHL* is involved in the circadian rhythm and the regulation of the timing of transition [63]; *PHYA* is involved in the regulation of flowering time and expression of its own gene as negative feedback [64]; *AGL62* is required for promoting the nuclear proliferation of early endosperm [65]; *AGL79* is involved in positively regulating the transition of the meristem from the vegetative to reproductive phase [66]; and *ATJ3* plays a

continuous role in plant development, such as in photoperiodism, flowering, and positive regulation of flower development [67]. Previous studies found that mRNAs (e.g., miR156, miR169 and miR172) play a crucial role in developmental processes in rice, wheat, and maize, especially in the formation of the floral meristem, with miR172 controlling *AP2-like* genes [23,68,69]. Studies on the flowering of Chenopodium quinoa found that pivotal flowering homologs, including photoreceptor genes *PHYA* and *CRY1*, as well as genes associated with florigen-encoding genes (*FT* and *TWIN SISTER of FT*) and circadian clock-related genes (*ELF3*, *LHY*, and *HY5*), were specifically affected by night-break and competed with the positive- and negative-flowering lncRNAs [70]. In this study, down-regulation of all the coexpressed mRNAs involved in circadian clock and meristem identity genes was observed, which can be considered acceptable and reasonable, because these mRNAs are often highly expressed at the plant development stage (at photoperiod), while their expression levels were examined during vernalization. In addition, increased expression with decreased temperatures indicates that their coexpressed lncRNAs play negative roles in regulating the flowering of *A. sinensis*.

The expression of numerous lncRNAs has been demonstrated to be significantly affected by various stresses [23,30]. During vernalization, plants have to face and adapt to low temperature [38]. To date, extensive studies have reported that lncRNAs participate in defense responses associated with plant immunity and adaptation to the environment [22]. Heat-responsive lncRNAs have been found to be differentially expressed in *Brassica juncea*, and cold-responsive lncRNAs have been identified in grape and *Arabidopsis* [71–73]. lncRNAs could regulate HSP family genes (*HSP82* and *HSP83*) in response to heat stress in *Populus x canadensis* Moench, and *HSP18.1* in response to Cd stress *Betula platyphylla* [74,75]. For the eight representative coexpressed mRNAs of the 14 lncRNAs linked with the temperature response, *ACBP6* confers resistance to cold and freezing [76]; *ENO2* acts as a positive regulator in response to cold stress [77]; *ADH1* is required for survival and acclimation in response to abiotic stresses (e.g., cold, salt, and drought) [78,79]; *CSP2* contributes to the enhancement of cold and freezing tolerance [80]; and *HSP17.8*, *HSP70-3*, *HSP70-10*, and *HSP90-3* play vital roles in adapting to biotic and abiotic stresses [81,82]. In this study, increased expression for cold-tolerated mRNAs (*ACBP6*, *ENO2*, *ADH1*, and *CSP2*), and decreased expression for heat-tolerated mRNAs (*HSP17.8*, *HSP70-3*, *HSP70-10*, and *HSP90-3*), were observed with decreased temperatures, which indicates that their coexpressed lncRNAs play positive roles in adapting to low temperatures.

For vernalization to occur, sources of energy (sugars) and carbohydrate metabolism are required [37]. In recent years, the roles of lncRNAs in regulating metabolism in cancer, insulin, and chicken have been reported [83–85], while, in plants, studies are still limited. For the five representative coexpressed mRNAs of the 19 lncRNAs linked with carbohydrate metabolism, *CSY4* is involved in the synthesis of socitrate from oxaloacetate [86]; *GAPCP2* plays a critical role in glycolysis and exhibits up-regulation under drought stress [87,88]; *At3g52990* is involved in the synthesis of pyruvate from D-glyceraldehyde 3-phosphate [89]; *PGM2* participates in the synthesis of glucose [90]; and *BAM1* is required for starch breakdown [91]. In this study, the differential expression of these coexpressed mRNAs regulating sucrose and starch metabolism provided energy for the morphogenesis of seedlings and adaptation to low temperatures during vernalization. Representatively, the decreased expression of metabolite-synthesized mRNAs (*CSY4*, *At3g52990*, and *PGM2*), and increased expression of energy-produced mRNA *GAPCP2* and metabolite-degraded mRNA *BAM1*, were observed with decreased temperatures, which indicates that their coexpressed lncRNAs play positive roles in carbohydrate metabolism.

Endogenous hormones such as gibberellin, auxin, cytokinin, brassinosteroid and abscisic acid can either inhibit or promote flowering [37]. In previous studies, a pre-miRNA of miR393 was identified in *Brassica rapa* during vernalization, and the overexpression of an miR393-resistant form of TIR1 (mTIR1) could enhance auxin sensitivity, thus leading to pleiotropic effects on plant development [92]. For the 8 representative coexpressed mRNAs of the 13 lncRNAs linked with hormone signaling, *ARF1* is involved in the recruitment

of COPI and GDAP1 to membranes and various auxin-dependent developmental processes [93]; *CUL1* participates in forming a SCF complex together with SKP1, RBX1, and a F-box protein and is involved in floral organ development, the auxin signaling pathway and ethylene signaling [94]; *SOFL4* and *SOFL5* are involved in cytokinin-mediated development [95]; *GRF2* and *GRF11* participate in the brassinosteroid (BR)-mediated signaling pathway [96,97]; *ERF3* is found to be differentially expressed in response to stresses and also regulates other ERFs [98]; and *SF1* is required for development and is involved in the alternative splicing of FLM pre-mRNA [99]. In this study, the differential expression of these mRNAs involved in hormone signaling also played certain roles in regulating the flower-bud differentiation of seedlings and cold tolerance during vernalization. In previous studies, *GRF2* and *ERF3* were found to be down-regulated, while *GRF11* was up-regulated in response to cold stress [100,101]; here, contrary findings for mRNAs *GRF2* and *GRF11* were observed with temperatures decreased, which indicate that their coexpressed lncRNAs may play negative roles in hormone signaling. The interaction between *SF1* and *FLM* pre-mRNA controls flowering time in response to temperature [102]; here, decreased expression of mRNA *SF1* was observed with decreased temperatures, which indicates that this coexpressed lncRNA may play a positive role in hormone signaling.

Energy generation and the transport of energy, nutrients and metabolites play essential roles in growth and development and stress tolerance [103]. For the 8 representative coexpressed mRNAs of the 43 lncRNAs linked with energy and transport, *PURU1* is involved in photorespiration and participates in preventing the excessive accumulation of 5-formyl tetrahydrofolate [104]; *MES16* is involved in chlorophyll breakdown by its demethylation [105]; *GPT1* is involved in NADPH generation through a series of processes including Glc6P transport, starch biosynthesis, fatty acid biosynthesis, and oxidative pentose phosphate [106]; *ABCF5* belongs to the ABC transporter superfamily and is involved in protein transport [107]; *SECA2* is involved in protein export coupling ATP hydrolysis [108]; *VPS26A* plays a role in vesicular protein sorting and is essential in endosome-to-Golgi retrograde transport [109]; *CML19* is a potential calcium sensor that binds calcium and is involved in the early response to stress [110]; and *NHX6* is involved in trafficking to the vacuole and exchanging the low-affinity electroneutral Na(+) or K(+)/H(+) [111]. In this study, the differential expression of these coexpressed mRNAs associated with energy and transport provided energy, nutrients, and metabolites for *A. sinensis* seedlings to obtain the capacity for vernalization and, meanwhile, to adapt to low temperatures.

Based on the above functional analysis of lncRNAs identified in this study, flowering pathways proposed in the previous literature [14–16] and a general model of stress-responsive regulation by regulatory lncRNAs [23], a model of vernalization-induced bolting and flowering by regulatory lncRNAs in *A. sinensis* is proposed (Figure 8). Briefly, the vernalization of seedlings firstly triggers the differential expression of lncRNAs; then, the lncRNAs either act as a precursor of miRNAs or as a miRNA target mimic, which further binds their related targets; then, the binding of targets regulates the expression of their downstream mRNAs that are involved in various biological processes, including the temperature response, flowering pathways (i.e., epigenetic modification, flowering induction, carbohydrate metabolism, and hormone signaling), as well as energy and transport; finally, these biological processes promote the transition of the meristem from the vegetative phase to the bolting and flowering of *A. sinensis*.

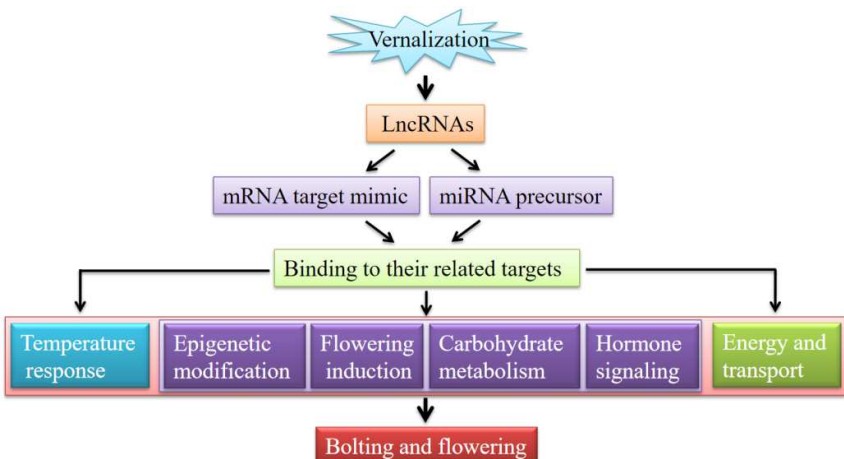

**Figure 8.** A proposed model of vernalization-induced bolting and flowering by regulatory lncRNAs in *A. sinensis*.

## 5. Conclusions

From the above observations, we found that the lncRNAs positively or negatively regulated the expression of their downstream mRNAs through epigenetic changes at the level of transcription and post-transcription for the flowering of *A. sinensis* during vernalization. This coexpressed mRNA analysis of lncRNAs focused on five pathways, namely (1) chromatin, DNA/RNA, and protein modification; (2) floral development; (3) temperature response; (4) carbohydrate metabolism; and (5) hormone signaling. While several candidate lncRNAs were identified, their causative roles require further investigations.

**Supplementary Materials:** The following supporting information can be downloaded at: https://www.mdpi.com/article/10.3390/cimb44050128/s1.

**Author Contributions:** X.L.: data curation, formal analysis, validation, and writing—original draft preparation; M.L. (Mimi Luo): formal analysis and validation; M.L. (Mengfei Li): conceptualization, project administration, supervision, and writing—review and editing; J.W.: funding acquisition and resources. All authors have read and agreed to the published version of the manuscript.

**Funding:** This research was funded by the State Key Laboratory of Aridland Crop Science/Gansu Agricultural University (GSCS-2021-Z03), National Natural Science Foundation of China (32160083), China Agriculture Research System of MOF and MARA (CARS-21), and Assurance Project of Ecological Planting and Quality of Daodi Herbs (202103003).

**Institutional Review Board Statement:** Not applicable.

**Informed Consent Statement:** Not applicable.

**Data Availability Statement:** The datasets are available at https://www.ncbi.nlm.nih.gov/bioproject/PRJNA789039 (release date on 13 January 2022).

**Conflicts of Interest:** All the authors declare no conflicts of interest.

## Abbreviations

| | |
|---|---|
| AP1 | APETALA 1 |
| CCS | Circular consensus sequence |
| CNCI | Coding–Noncoding Index |
| COLDAIR | COLD-ASSISTED INTRONIC NON-CODING RNA |
| COOLAIR | COLD-INDUCED LONG ANTISENSE INTRAGENIC RNAs |
| CPC | Coding-Potential Calculator |
| FLC | FLOWERING LOCUS C |
| FLM | FLOWERING LOCUS M |
| FLNC | Full-length nonchimeric |

| FT | FLOWERING LOCUS T |
| GA | Gibberellin |
| GA2OX1 | Gibberellin 2-β-dioxygenase 1 |
| GO | Gene Ontology |
| KEGG | Kyoto Encyclopedia of Genes and Genomes |
| KOG | euKaryotic orthologous groups of proteins |
| lncRNAs | Long noncoding RNAs |
| NCBI | National Center for Biotechnology Information |
| ncRNAs | Noncoding RNAs |
| NR | NCBI nonredundant protein |
| PHYA | PHYOCHROME A |
| REL | Relative expression level |
| SOC1 | SUPPRESSOR OF OVEREXPRESSION OF CONSTANS 1 |
| sRNAs | Small RNAs |
| TAIR | The Arabidopsis Information Resource |

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
