# Peer review of "Transcriptomic Analysis Reveals LncRNAs Associated with Flowering of Angelica sinensis during Vernalization"

_cimb, doi:10.3390/cimb44050128_

Round 1
Reviewer 1 Report
In this manuscript, the author did the transcriptomic analysis of LncRNAs associated with the flowering of Angelica sinensis during vernalization. In this study, author lncRNAs associated with flowering were identified based on a full-length transcriptomic analysis of A. sinensis at vernalization and freezing temperatures, and the co-expressed mRNAs of lncRNAs were validated by qRT-PCR. We found that a total 2327 lncRNAs were obtained after assessing the protein-coding potential of co-expressed mRNAs, with 607 lncRNAs aligned against the TAIR database of the model plant Arabidopsis 345 lncRNAs identified and 272 lncRNAs characterized on the SwissProt database. Based on the biological functions of co-expressed mRNAs, the 272 lncRNAs were divided into six categories: (1) chromatin, DNA/RNA and protein modification; (2) flowering; (3) stress response; (4) metabolism; (5) bio-signaling; and (6) energy and transport. The differential expression levels of representatively co-expressed mRNAs were almost consistent with the flowering of A. sinensis. It can be concluded that the flowering of A. sinensis is positively or negatively regulated by the lncRNAs, which will provide new insights into the regulation mechanism of the flowering of A. sinensis.
Manuscripts need English editing, and it would be nice if authors tried to validate the function of at least one gene identified in this study. A hypothetical figure that depicts the finding of this study will be good.
I found plagiarism in this manuscript at L32-33, L53-55, L60-61, L63-65, L67-69, L89-90, L93-94, L95-97, L100-102, L104, L108-111, L118-120, L250-252, L260, L270, L279-271, L321-322, L325-326, L334-335, L355-357, L364-367, L374-379, L385-386, and L392-393. Please clean it.
Author Response
Many thanks for your and reviewer’s comments that are helpful to improve our paper much better now. We have tried to address and correct each comment. Attachments below with our responses are shown in bold. Revised parts are highlighted in red in the manuscript.
1: Manuscripts need English editing, and it would be nice if authors tried to validate the function of at least one gene identified in this study. A hypothetical figure that depicts the finding of this study will be good.
Thanks very much for your suggestion, the manuscript has been English edited by MDPI Language Editing Services (ID: 43227).
According to your comments, Figure 8 has been proposed to depict the finding of this study. The description has also been added in the text: “Based on the above functional analysis of lncRNAs identified in this study, flowering pathways proposed in the previous literature [14-16] and a general model of stress-responsive regulation by regulatory lncRNAs [23], a model of vernalization-induced bolting and flowering by regulatory lncRNAs in A. sinensis is proposed (Figure 8). Briefly, the vernalization of seedlings will firstly trigger the differential expression of lncRNAs; then, the lncRNAs will either act as a precursor of miRNAs or as a miRNA target mimic, which will further bind their related targets; then, the binding of targets will regulate the expression of their downstream mRNAs that are involved in various biological processes, including the temperature response, flowering pathways (i.e., epigenetic modification, flowering induction, carbohydrate metabolism and hormone signaling), as well as energy and transport; finally, these biological processes will promote the transition of the meristem from the vegetative phase to the bolting and flowering of A. sinensis.” (Page 15, lines 410-422)
Indeed, it would be nice to validate the function of genes by virus induced gene silencing (VIGS) or CRISPR-Cas9, which will be further performed in our next experiments. After all, the pandemic of Covid-19 has been delaying our research process and restricting our researchers to the Lab for a long time. We hope that the reviewers can understand the current manuscript without validating the function of at least one gene by VIGS.
2: I found plagiarism in this manuscript at L32-33, L53-55, L60-61, L63-65, L67-69, L89-90, L93-94, L95-97, L100-102, L104, L108-111, L118-120, L250-252, L260, L270, L279-271, L321-322, L325-326, L334-335, L355-357, L364-367, L374-379, L385-386, and L392-393. Please clean it.
Thanks for your kind reminder. We have reconstructed these sentences to avoid the overlaps and plagiarism throughout the text.
Reviewer 2 Report
The article presents transcriptomic analysis of lncRNA based on full-length transcriptomic analysis of A. sinensis at vernalization and freezing temperatures, as well as the co-expressed mRNAs of 49 lncRNAs, validated by qRT-PCR, devided into 6 group: chromatin, DNA/RNA and protein modification (I), flowering induction (II), stress response (III), metabolism (IV), bio-signaling (V), and energy and transport (VI). This work is very interesting cognitively and important from the economic point of view, however, there is some details that should be added:
Materials and methods:
117-118: “…41 primer sequences of representatively co-expressed mRNAs…” – I assumed, that it should be probably “…49 pair of primers of representatively co-expressed mRNAs…” The qPCR conditions for all pairs of primers should be described.
The authors provided the information on the collection of three biological samples for transcriptomic analysis and qRT-PCR validation. Three biological samples of each variant tested the minimum number for the qPCR, and at least three technical replication should be prepared for each biological sample. There is no information how many technical samples were prepared. Only if the standard deviation in these case is low, it can be accepted, otherwise it is recommended to use more trials to minimize the error.
Additionally, there is no information about the standard curve. It allows you to assess the quantitative effectiveness of the PCR test and its dynamic range as well as the limit of detection and the limit of quantification, in accordance with the latest recommendations (eg.: Svec et al., 2015).
The R2 values and PCR efficiency obtained for the standard curves for the reference genes as well as investigated ones should be added to the supplementary material and the obtained expression values should be corrected using efficiency.
Results:
Based on biological functions, 272 characterized lncRNAs were divided into six categories: chromatin, DNA/RNA and protein modification (29), flowering (36), stress response (24), metabolism (117), bio-signaling (23), and energy and transport (43). For the qRT-PCR analysis the authors have chosen “representative” co-expressed mRNAs from all of mentioned group. What does exactly mean “representative”? How the lncRNAs tested were selected?
The expression level as well as standard deviation for every variant tested should be added in supplementary file.
Discussion:
The Discussion section provided a lot of information about the investigated lncRNAs and molecular mechanism in which they are involved, and not much information about the results obtained in this experiment as well as the comparison with other. Some of the information may be included in the introduction section.
Author Response
Many thanks for your and reviewer’s comments that are helpful to improve our paper much better now. We have tried to address and correct each comment. Attachments below with our responses are shown in bold. Revised parts are highlighted in red in the manuscript.
1>Materials and methods:
1: 117-118: “…41 primer sequences of representatively co-expressed mRNAs…” – I assumed, that it should be probably “…49 pair of primers of representatively co-expressed mRNAs…” The qPCR conditions for all pairs of primers should be described.
Thanks very much for kind review. Firstly, the number “41” that was written by mistake has been revised to “49”. Secondly, the qRT-PCR conditions has been described in the text: “First-strand cDNA synthesis and qRT-PCR reaction were carried out using SuperRealPreMix Plus (FP205; Tiangen Biotech., Beijing, China) according to the manufacturer's instructions; specifically, the cDNA was synthesized successively with one cycle (95°C, 15 min) and 40 cycles (95°C, 10 s; 60°C, 20 s; and 72°C, 30 s), and the qRT-PCR reaction was incubated successively at 95°C for 15 s, 60°C for 1 min and 95°C for 1 s.” (Page 3, lines 115-120)
2: The authors provided the information on the collection of three biological samples for transcriptomic analysis and qRT-PCR validation. Three biological samples of each variant tested the minimum number for the qPCR, and at least three technical replication should be prepared for each biological sample. There is no information how many technical samples were prepared. Only if the standard deviation in these case is low, it can be accepted, otherwise it is recommended to use more trials to minimize the error.
In this study, three biological along with three technical replications were performed, and the information has been added in “2.6. Statistical Analysis”: “In order to obtain the precise estimation of PCR efficiency, each experiment for qRT-PCR validation was performed with three biological replicates, along with three technical replicates [36].” (Page 3, lines 134-136)
From the values that were presented in Figures 2 to Figures 7, the standard deviations for all the relative expression levels of the 49 candidate genes are very low; meanwhile, their standard deviations have been added to supplementary material (Table S1).
3: Additionally, there is no information about the standard curve. It allows you to assess the quantitative effectiveness of the PCR test and its dynamic range as well as the limit of detection and the limit of quantification, in accordance with the latest recommendations (eg.: Svec et al., 2015).
In this study, there are 50 genes (i.e., 49 candidate genes and actin gene) used for qRT-PCR test. Herein, only the actin gene was performed for the standard curve, after all, the actin gene was used as a reference control gene to calculate the relative expression level of co-expressed mRNAs using a 2-△△Ct method.
4: The R2 values and PCR efficiency obtained for the standard curves for the reference genes as well as investigated ones should be added to the supplementary material and the obtained expression values should be corrected using efficiency.
According to your comments, the standard curve of actin gene along with the R2 value and PCR efficiency has been provided as Figure S3, and the information has been added in the text: “Herein, the cycle threshold (Ct) values and standard curves of the ACT gene at different volumes (0.25, 0.5, 1.0, 1.5, 2.0 and 3.0 μL) was built to correct the gene expression level (Figure S3 and Figure S4)”. (Page 3, lines 122-124)
Additionally, the expression values of 49 representative genes have been provided as Table S1.
2>Results:
1: Based on biological functions, 272 characterized lncRNAs were divided into six categories: chromatin, DNA/RNA and protein modification (29), flowering (36), stress response (24), metabolism (117), bio-signaling (23), and energy and transport (43). For the qRT-PCR analysis the authors have chosen “representative” co-expressed mRNAs from all of mentioned group. What does exactly mean “representative”? How the lncRNAs tested were selected?
As described in the Introduction section (Page 2, lines 47-54), four main pathways are involved in flowering including: photoperiod, vernalization, carbohydrate metabolism, and hormone signaling, based on previously published literatures.
In this study, among of the 272 lncRNAs, 87 co-expressed mRNAs were found to be directly linked with the four flowering pathways and temperature response.
Firstly, we randomly selected 41 of 87 co-expressed mRNAs that are directly linked with chromatin, DNA/RNA and protein modification (14/29), flowering (6/12), temperature response (8/14), carbohydrate metabolism (5/19); and hormone signaling (8/13). Herein, the mRNAs linked with temperature response were selected because the seedlings were treated at low temperatures, in other words, in response to vernalization and freezing temperatures (avoided vernalization).
Secondly, we randomly selected 8 of 43 co-expressed mRNAs that are directly linked with energy and transport, which are indirectly involved in the four flowering pathways.
2: The expression level as well as standard deviation for every variant tested should be added in supplementary file.
According to your comments, the expression level and standard deviation for every variant tested have been added to supplementary material. Meanwhile, the information has been added in the text: “and the expression levels of the 49 candidate genes and their standard deviations for every variant have been added to the Supplementary Material (Table S1)” (Page 3, lines 124-126)
3>Discussion:
1: The Discussion section provided a lot of information about the investigated lncRNAs and molecular mechanism in which they are involved, and not much information about the results obtained in this experiment as well as the comparison with other. Some of the information may be included in the introduction section.
According to your and other reviewer’s comments, the discussion has been improved by comparing the results in this study with previously published literatures.
Reviewer 3 Report
In the manuscript entitled “Transcriptomic Analysis Reveals LncRNAs Associated with Flowering of Angelica sinensis during Vernalization” the authors identified the lncRNAs associated with flowering in A. sinensis at vernalization and freezing temperatures, and also performed the co-expressed mRNAs of lncRNAs. Overall, this study provides excellent information that could be used in future studies. However, some issues need to be resolved (given below) before its final acceptance.
- The lncRNAs also function through miRNAs for transcriptional, post-transcriptional and epigenetic gene regulation through diverse molecular mechanisms. The author should also explore and identify the lncRNAs acting as precursors and target mimic of known miRNAs.
- In Figures 2,3,4,5,6,7 related expression levels of co-expressed mRNAs of lncRNAs linked. A statistical test should be performed to show the significant difference in genes expressions.
- Validation of selected lncRNA should be conducted using QRT-PCR.
- Since the finding in this paper was interesting, but discussion of the results needs improvement. In my opinion, they have to discuss their results and compared them with earlier and recently published papers in more depth.
Author Response
Many thanks for your and reviewer’s comments that are helpful to improve our paper much better now. We have tried to address and correct each comment. Attachments below with our responses are shown in bold. Revised parts are highlighted in red in the manuscript.
1: The lncRNAs also function through miRNAs for transcriptional, post-transcriptional and epigenetic gene regulation through diverse molecular mechanisms. The author should also explore and identify the lncRNAs acting as precursors and target mimic of known miRNAs.
Thanks for your suggestion, the description involved in “lncRNAs acting as precursors and target mimic of known miRNAs” has been added in the Introduction section.
In the Introduction section, the roles of miRNAs and lncRNAs have been added: “Both miRNAs and lncRNAs can influence plant developmental processes and stress responses [20], with the former being negative regulators functioning as specificity determinants, or guides, within complexes that promote the degradation of mRNA targets, and the latter acting either as precursors of miRNAs or endogenous target mimics (TMs), which mimic the real targets of miRNAs, thus rendering the corresponding miRNAs ineffective [21]. For example, previous studies on resistance against leaf rust in wheat found that 50 miRNAs and 1178 lncRNAs were identified and 49 lncRNAs were found to be the targets for miRNAs, with 1 lncRNA acting as a precursor of 2 miRNAs and 3 lncRNAs acting as TMs [22].” (Page 2, lines 60-68)
2: In Figures 2, 3, 4, 5, 6, 7 related expression levels of co-expressed mRNAs of lncRNAs linked. A statistical test should be performed to show the significant difference in genes expressions.
According to your comments, the significant differences in Figure 2 to Figure 7 have been added.
3: Validation of selected lncRNA should be conducted using QRT-PCR.
Indeed, the co-expressed mRNAs of lncRNAs were validated by qRT-PCR, and the information has been provided in the “2. Materials and Methods——2.5. qRT-PCR Validation”. (Page 3, lines 112-132)
4: Since the finding in this paper was interesting, but discussion of the results needs improvement. In my opinion, they have to discuss their results and compared them with earlier and recently published papers in more depth.
According to your and other reviewer’s comments, the discussion has been improved by comparing the results in this study with previously published literatures.
Round 2
Reviewer 1 Report
I am happy with the authors reply and the manuscript can be accepted in its current format.
Reviewer 2 Report
I have no further comments, manuscript can be published in present form.
Reviewer 3 Report
The authors have clarified most of the questions I raised in my previous review. Now I feel the manuscript could be accepted for publication.